# TARGETED ADVERSARIAL EXAMPLES FOR BLACK BOX AUDIO SYSTEMS

## ABSTRACT

The application of deep recurrent networks to audio transcription has led to impressive gains in automatic speech recognition (ASR) systems. Many have demonstrated that small adversarial perturbations can fool deep neural networks into incorrectly predicting a specified target with high confidence. Current work on fooling ASR systems have focused on white-box attacks, in which the model architecture and parameters are known. In this paper, we adopt a black-box approach to adversarial generation, combining the approaches of both genetic algorithms and gradient estimation to solve the task. We achieve a 89.25% targeted attack similarity after 3000 generations while maintaining 94.6% audio file similarity.

## 1 INTRODUCTION

Although neural networks have incredible expressive capacity, which allow them to be well suited for a variety of machine learning tasks, they have been shown to be vulnerable to adversarial attacks over multiple network architectures and datasets Goodfellow et al. (2014). These attacks can be done by adding small perturbations to the original input so that the network misclassifies the input but a human does not notice the difference.

So far, there has been much more work done in generating adversarial examples for image inputs than for other domains, such as audio. Voice control systems are widely used in many products from personal assistants, like Amazon Alexa and Apple Siri, to voice command technologies in cars. One main challenge for such systems is determining exactly what the user is saying and correctly interpreting the statement. As deep learning helps these systems better understand the user, one potential issue is targeted adversarial attacks on the system, which perturb the waveform of what the user says to the system to cause the system to behave in a predetermined inappropriate way. For example, a seemingly benign TV advertisement could be adversely perturbed to cause Alexa to interpret the audio as "Alexa, buy 100 headphones." If the original user went back to listen to the audio clip that prompted the order, the noise would be almost undetectable to the human ear.

There are multiple different methods of performing adversarial attacks depending on what information the attacker has about the network. If given access to the parameters of a network, white box attacks are most successful, such as the Fast Gradient Sign Method Goodfellow et al. (2014) or DeepFool Moosavi-Dezfooli et al. (2015). However, assuming an attacker has access to all the parameters of a network is unrealistic in practice. In a black box setting, when an attacker only has access to the logits or outputs of a network, it is much harder to consistently create successful adversarial attacks.

In certain special black box settings, white box attack methods can be reused if an attacker creates a model that approximates the original targeted model. However, even though attacks can transfer across networks for some domains, this requires more knowledge of how to solve the task that the original model is solving than an attacker may have Liu et al. (2016); Papernot et al. (2016). Instead, we propose a novel combination of genetic algorithms and gradient estimation to solve this task. The first phase of the attack is carried out by genetic algorithms, which are a gradient-free method of optimization that iterate over populations of candidates until a suitable sample is produced. In order to limit excess mutations and thus excess noise, we improve the standard genetic algorithm with a new momentum mutation update. The second phase of the attack utilizes gradient estimation, where the gradients of individual audio points are estimated, thus allowing for more careful noise placement

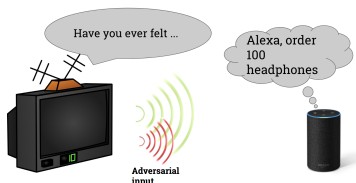

Figure 1: Example of targeted adversarial attack on speech to text systems in practice

when the adversarial example is nearing its target. The combination of these two approaches provides a 89.25% average targeted attack similarity with a 94.6% audio file similarity after 3000 generations.

## 1.1 PROBLEM STATEMENT

Adversarial attacks can be created given a variety of information about the neural network, such as the loss function or the output probabilities. However in a natural setting, usually the neural network behind such a voice control system will not be publicly released so an adversary will only have access to an API which provides the text the system interprets given a continuous waveform. Given this constraint, we use the open sourced Mozilla DeepSpeech implementation as a black box system, without using any information on how the transcription is done.

We perform our black box targeted attack on a model $M$ given a benign input $x$ and a target $t$ by perturbing $x$ to form the adversarial input $x' = x + \delta$, such that $M(x') = t$. To minimize the audible noise added to the input, so a human cannot notice the target, we maximize the cross correlation between $x$ and $x'$. A sufficient value of $\delta$ is determined using our novel black box approach, so we do not need access to the gradients of $M$ to perform the attack.

## 1.2 PRIOR WORK

Compared to images, audio presents a much more significant challenge for models to deal with. While convolutional networks can operate directly on the pixel values of images, ASR systems typically require heavy pre-processing of the input audio. Most commonly, the Mel-Frequency Cepstrum (MFC) transform, essentially a fourier transform of the sampled audio file, is used to convert the input audio into a spectogram which shows frequencies over time. Models such as DeepSpeech (Fig. 2) use this spectogram as the initial input.

In a foundational study for adversarial attacks, Cisse et al. (2017) developed a general attack framework to work across a wide variety of models including images and audio. When applying their method to audio samples, they ran into the roadblock of backpropagating through the MFC conversion layer. Thus, they were able to produce adversarial spectograms but not adversarial .wav files.

Carlini & Wagner (2018) overcame this challenge by developing a method of passing gradients through the MFC layer, a task which was previously proved to be difficult Cisse et al. (2017). They applied their method to the Mozilla DeepSpeech model, which is a complex, recurrent, character-level network that can decode translations at up to 50 characters per second. With a gradient connection all the way to the raw input, they were able to achieve impressive results, including generating samples over 99.9% similar with a targeted attack accuracy of 100%. While the success of this attack opens new doors for white box attacks, adversaries in a real-life setting commonly do not have knowledge of model architectures or parameters.

Alzantot et al. (2018) have demonstrated that black-box approaches for targeted attacks on ASR systems are possible. Using a genetic algorithm approach, they were able to iteratively apply noise to audio samples, pruning away poor performers at each generation, and ultimately end up with a perturbed version of the input that successfully fooled a classification system. This attack was conducted on the Speech Commands classification model Alzantot et al. (2018), which is a lightweight convolutional model for classifying up to 50 different single-word phrases.

Extending the research done by Alzantot et al. (2018), we propose a genetic algorithm and gradient estimation approach to create targeted adversarial audio, but on the more complex DeepSpeech system. The difficulty of this task comes in attempting to apply black-box optimization to a deeply-

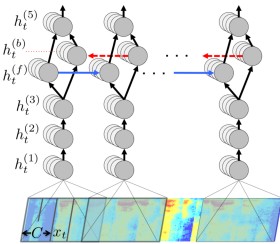

Figure 2: Diagram of Baidu's DeepSpeech model Hannun et al. (2014)

layered, highly nonlinear decoder model that has the ability to decode phrases of arbitrary length. Nevertheless, the combination of two differing approaches as well as the momentum mutation update bring new success to this task.

### 1.3 BACKGROUND

**Dataset**   For the attack, we follow Carlini & Wagner (2018) and take the first 100 audio samples from the CommonVoice test set. For each, we randomly generate a 2-word target phrase and apply our black-box approach to construct an adversarial example. More details on evaluation can be found in section 3. Each sample in the dataset is a .wav file, which can easily be deserialized into a numpy array. Our algorithm operates directly on the numpy arrays, thus bypassing the difficulty of dealing with the MFC conversion.

**Victim model**   The model we attack is Baidu's DeepSpeech model Hannun et al. (2014), implemented in Tensorflow and open-sourced by Mozilla.[1] Though we have access to the full model, we treat it as if in a black box setting and only access the output logits of the model. In line with other speech to text systems Chiu et al. (2017); Cisse et al. (2017), DeepSpeech accepts a spectrogram of the audio file. After performing the MFC conversion, the model consists 3 layers of convolutions, followed by a bi-directional LSTM, followed by a fully connected layer. This layer is then fed into the decoder RNN, which outputs logits over the distribution of output characters, up to 50 characters per second of audio. The model is illustrated in figure 2.

**Connectionist temporal classication**   While the DeepSpeech model is designed to allow arbitrary length translations, there is no given labeled alignment of the output and input sequences during training time. Thus the connectionist temporal classication loss (CTC Loss) is introduced, as it allows computing a loss even when the position of a decoded word in the original audio is unknown Carlini & Wagner (2018).

DeepSpeech outputs a probability distribution over all characters at every frame, for 50 frames per second of audio. In addition to outputting the normal alphabet a-z and space, it can output special character $\epsilon$. Then CTC decoder $C(\cdot)$ decodes the logits as such: for every frame, take the character with the max logit. Then first, remove all adjacent duplicate characters, and then second, remove any special $\epsilon$ characters. Thus $aab\epsilon\epsilon b$ will decode to $abb$ Carlini & Wagner (2018).

As we can see, multiple outputs can decode to the same phrase. Following the notation in Carlini & Wagner (2018), for any target phrase $p$, we call $\pi$ an alignment of $p$ if $C(\pi) = p$. Let us also call the output distribution of our model $y$. Now, in order to find the likelihood of alignment $\pi$ under $y$:

$$Pr(p|y) = \sum_{\pi | C(\pi) = p} Pr(\pi | y) = \sum_{\pi | C(\pi) = p} \prod_i y_\pi^i$$

as noted by Carlini & Wagner (2018). This is the objective we use when scoring samples from the populations in each generation of our genetic algorithm as well as the score used in estimating gradients.

---

[1] https://github.com/mozilla/DeepSpeech

**Greedy decoding**   As in traditional recurrent decoder systems, DeepSpeech typically uses a beam search of beam width 500. At each frame of decoding, 500 of the most likely $\pi$ will be evaluated, each producing another 500 candidates for a total of 2500, which are pruned back down to 500 for the next timestep. Evaluating multiple assignments this way increases the robustness of the model decoding. However, following work in Carlini & Wagner (2018), we set the model to use greedy decoding. At each timestep only 1 $\pi$ is evaluated, leading to a greedy assignment:

$$decode(x) = C(\arg\max_{\pi} Pr(y(x)|\pi))$$

Thus, our genetic algorithm will focus on creating perturbations to the most likely sequence (if greedily approximated).

## 2   BLACK BOX ATTACK ALGORITHM

---
**Algorithm 1** Black box algorithm for generating adversarial audio sample

---
**Input:** Original benign input $x$ Target phrase $t$
**Output:** Adversarial Audio Sample $x'$
  population $\leftarrow [x] * populationSize$
  **while** iter $< maxIters$ and $Decode$(best)$! = t$ **do**
    scores $\leftarrow -CTCLoss$(population, $t$)
    best $\leftarrow$ population$[Argmax$(scores$)]$

    **if** $EditDistance(t, Decode$(best)$) > 2$ **then**
      // phase 1 - do genetic algorithm
      **while** populationSize children have not been made **do**
        Select $parent1$ from $topk$(population) according to $softmax$(their score)
        Select $parent2$ from $topk$(population) according to $softmax$(their score)
        child $\leftarrow Mutate(Crossover$(parent1, parent2), p)
      **end while**
      newScores $\leftarrow -CTCLoss$(newPopulation, $t$)
      p $\leftarrow MomentumUpdate$(p, newScores, scores)

    **else**
      // phase 2 - do gradient estimation
      top-element $\leftarrow top$(population)
      grad-pop $\leftarrow n$ copies of top-element, each mutated slightly at one index
      grad $\leftarrow (-CTCLoss$(grad-pop) $-$ scores)/mutation-delta
      pop $\leftarrow$ top-element $+$ grad
    **end if**
  **end while**
  **return**  best

---

### 2.1   GENETIC ALGORITHM

As mentioned previously, Alzantot et al. (2018) demonstrated the success of a black-box adversarial attack on speech-to-text systems using a standard genetic algorithm. The basic premise of our algorithm is that it takes in the benign audio sample and, through trial and error, adds noise to the sample such that the perturbed adversarial audio is similar to the benign input yet is decoded as the target, as shown in Figure 3. A genetic algorithm works well for a problem of this nature because it is completely independent of the gradients of the model. Alzantot et al. (2018) used a limited dataset consisting of audio samples with just one word and a classification with a predefined number of classes. In order to extend this algorithm to work with phrases and sentences, as well as with CTC Loss, we make modifications to the genetic algorithm and introduce our novel momentum mutation.

The genetic algorithm works by improving on each iteration, or generation, through evolutionary methods such as Crossover and Mutation Holland (1992). For each iteration, we compute the score

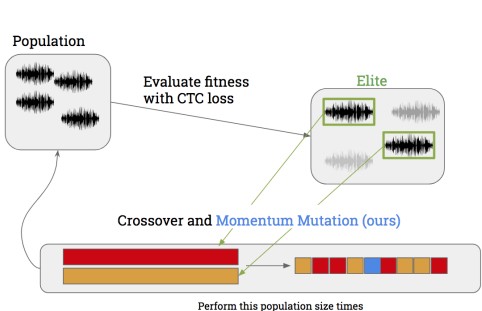

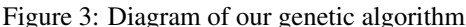

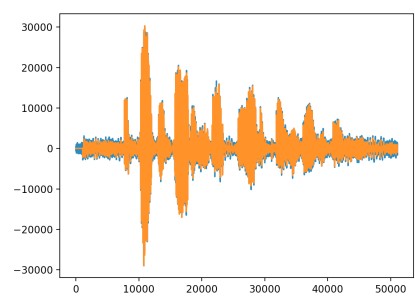

Figure 4: Overlapping of adversarial (blue) and original (orange) audio sample waveforms. The perturbation is barely noticeable

Figure 3: Diagram of our genetic algorithm

for each sample in the population to determine which samples are the best. Our scoring function was the CTC-Loss, which as mentioned previously, is used to determine the similarity between an input audio sequence and a given phrase. We then form our elite population by selecting the best scoring samples from our population. The elite population contains samples with desirable traits that we want to carry over into future generations. We then select parents from the elite population and perform Crossover, which creates a child by taking around half of the elements from $parent1$ and the other half from $parent2$. The probability that we select a sample as a parent is a function of the sample's score. With some probability, we then add a mutation to our new child. Finally, we update our mutation probabilities according to our momentum update, and move to the next iteration. The population will continue to improve over time as only the best traits of the previous generations as well as the best mutations will remain. Eventually, either the algorithm will reach the max number of iterations, or one of the samples is exactly decoded as the target, and the best sample is returned.

## 2.2 MOMENTUM MUTATION

---

**Algorithm 2** Mutation

**Input:** Audio Sample $x$
   Mutation Probability $p$
**Output:** Mutated Audio Sample $x'$

    **for all** $e$ in $x$ **do**
       noise $\leftarrow Sample(\mathcal{N}(\mu, \sigma^2))$
       **if** $Sample(\text{Unif}(0, 1)) < p$ **then**
          $e' \leftarrow e + filter_{highpass}(\text{noise})$
       **end if**
    **end for**
    **return** $x'$

---

The mutation step is arguably the most crucial component of the genetic algorithm and is our only source of noise in the algorithm. In the mutation step, with some probability, we randomly add noise to our sample. Random mutations are critical because it may cause a trait to appear that is beneficial for the population, which can then be proliferated through crossover. Without mutation, very similar samples will start to appear across generations; thus, the way out of this local maximum is to nudge it in a different direction in order to reach higher scores.

Furthermore, since this noise is perceived as background noise, we apply a filter to the noise before adding it onto the audio sample. The audio is sampled at a rate of $f_s = 16kHz$, which means that the maximum frequency response $f_{max} = 8kHz$. As seen by Reichenbach & Hudspeth (2012), given that the human ear is more sensitive to lower frequencies than higher ones, we apply a highpass filter

at a cutoff frequency of $f_{cutoff} = 7kHz$. This limits the noise to only being in the high-frequency range, which is less audible and thus less detectable by the human ear.

While mutation helps the algorithm overcome local maxima, the effect of mutation is limited by the *mutation probability*. Much like the step size in SGD, a low mutation probability may not provide enough randomness to get past a local maximum. If mutations are rare, they are very unlikely to occur in sequence and *add on* to each other. Therefore, while a mutation might be beneficial when accumulated with other mutations, due to the low mutation probability, it is deemed as not beneficial by the algorithm in the short term, and will disappear within a few iterations. This parallels the step size in SGD, because a small step size will eventually converge back at the local minimum/maximum. However, too large of a mutation probability, or step size, will add an excess of variability and prevent the algorithm from finding the global maximum/minimum. To combat these issues, we propose **Momentum Mutation**, which is inspired by the Momentum Update for Gradient Descent. With this update, our mutation probability changes in each iteration according to the following exponentially weighted moving average update:

$$p_{new} = \alpha \times p_{old} + \frac{\beta}{|currScore - prevScore|}$$

With this update equation, the probability of a mutation increases as our population fails to adapt meaning the current score is close to the previous score. The momentum update adds acceleration to the mutation probability, allowing mutations to accumulate and add onto each other by keeping the mutation probability high when the algorithm is stuck at a local maximum. By using a moving average, the mutation probability becomes a smooth function and is less susceptible to outliers in the population. While the momentum update may overshoot the target phrase by adding random noise, overall it converges faster than a constant mutation probability by allowing for more acceleration in the right directions.

---

**Algorithm 3** Momentum Mutation Update

---

**Input:** Mutation Probability $p$
    Scores for the new population $newScores$
    Scores for the previous population $scores$
**Output:** Updated mutation probability $p_{new}$

    currScore $= max(\text{newScores})$
    prevScore $= max(\text{scores})$
    $p_{new} = \alpha \times p_{old} + \frac{\beta}{|currScore-prevScore|}$
    **return** $p_{new}$

---

### 2.3 Gradient estimation

Genetic algorithms work well when the target space is large and a relatively large number of mutation directions are potentially beneficial; the strength of these algorithms lies in being able to search large amounts of space efficiently Godefroi & Khurshid (unkown). When an adversarial sample nears its target perturbation, this strength of genetic algorithms turn into a weakness, however. Close to the end, adversarial audio samples only need a few perturbations in a few key areas to get the correct decoding. In this case, gradient estimation techniques tend to be more effective. Specifically, when edit distance of the current decoding and the target decoding drops below some threshold, we switch to phase 2. When approximating the gradient of a black box system, we can use the technique proposed by Nitin Bhagoji et al. (2017):

$$FD_x(x, \delta) = \begin{bmatrix} (g(x + \delta_1) - g(x))/\delta \\ \vdots \\ (g(x + \delta_n) - g(x))/\delta \end{bmatrix}$$

Here, $x$ refers to the vector of inputs representing the audio file. $\delta_i$ refers to a vector of all zeros, except at the $i^{th}$ position in which the value is a small $\delta$. $g(\cdot)$ represents the evaluation function,

which in our case is CTCLoss. Essentially, we are performing a small perturbation at each index and individually seeing what the difference in CTCLoss would be, allowing us to compute a gradient estimate with respect to the input $x$.

However, performing this calculation in full would be prohibitively expensive, as the audio is sampled at $16kHz$ and so a simple 5-second clip would require 80,000 queries to the model for just one gradient evaluation! Thus, we only randomly sample 100 indices to perturb each generation when using this method. When the adversarial example is already near the goal, gradient estimation makes the tradeoff for more informed perturbations in exchange for higher compute.

## 3 EVALUATION

### 3.1 METRICS

We tested our algorithm by running it on a 100 sample subset of the Common Voice dataset. For each audio sample, we generated a single random target phrase by selecting two words uniformly without replacement from the set of 1000 most common words in the English language. The algorithm was then run for each audio sample and target phrase pair for 3000 generations to produce a single adversarial audio sample.

We evaluated the performance of our algorithm in two primary ways. The first method is determining the accuracy with which the adversarial audio sample gets decoded to the desired target phrase. For this, we use the Levenshtein distance, or the minimum character edit distance, between the desired target phrase and the decoded phrase as the metric of choice. We then calculated the percent similarity between the desired target and the decoded phrase by calculating the ratio of the Levenshtein distance and the character length of the original input, ie. $1 - \frac{Levenshtein(M(x'),t)}{len(M(x))}$. The second method is determining the similarity between the original audio sample and the adversarial audio sample. For this, we use the accepted metric of the cross correlation coefficient between the two audio samples.

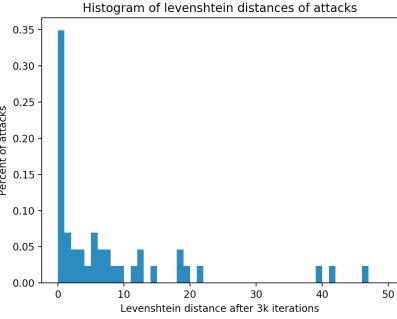

Figure 5: Histogram of levenshtein distances of attacks.

### 3.2 RESULTS

Of the audio samples for which we ran our algorithm on, we achieved a 89.25% similarity between the final decoded phrase and the target using Levenshtein distance, with an average of 94.6% correlation similarity between the final adversarial sample and the original sample. The average final Levenshtein distance after 3000 iterations is 2.3, with 35% of the adversarial samples achieving an exact decoding in less than 3000 generations, and 22% of the adversarial samples achieving an exact decoding in less than 1000 generations.

One thing to note is that our algorithm was 35% successful in getting the decoded phrase to match the target exactly; however, noting from figure 5, the vast majority of failure cases are only a few edit distances away from the target. This suggests that running the algorithm for a few more iterations could produce a higher success rate, although at the cost of correlation similarity. Indeed, it becomes apparent that there is a tradeoff between success rate and audio similarity such that this threshold could be altered for the attacker's needs.

A comparison of white box targeted attacks, black box targeted attacks on single words (classification), and our method:

| Metric | White Box Attacks | Our Method | Single Word Black Box |
|---|---|---|---|
| Targeted attack success rate | 100% | 35% | 87% |
| Average similarity score | 99.9% | 94.6% | 89% |
| Similarity score method | cross-correlation | cross-correlation | human study |
| Loss used for attack | CTC | CTC | Softmax |
| Dataset tested on | Common Voice | Common Voice | Speech Commands |
| Target phrase generation | Single sentence | Two word phrases | Single word |

One helpful visualization of the similarity between the original audio sample and the adversarial audio sample through the overlapping of both waveforms, as shown in figure 4. As the visualization shows, the audio is largely unchanged, and the majority of the changes to the audio is in the relatively low volume noise applied uniformly around the audio sample. This results in an audio sample that still appears to transcribe to the original intended phrase when heard by humans, but is decoded as the target adversarial phrase by the DeepSpeech model.

That 35% of random attacks were successful in this respect highlights the fact that black box adversarial attacks are definitely possible and highly effective at the same time.

## 4 CONCLUSION

In combining genetic algorithms and gradient estimation we are able to achieve a black box adversarial example for audio that produces better samples than each algorithm would produce individually. By initially using a genetic algorithm as a means of exploring more space through encouragement of random mutations and ending with a more guided search with gradient estimation, we are not only able to achieve perfect or near-perfect target transcriptions on most of the audio samples, we were able to do so while retaining a high degree of similarity. While this remains largely as a proof-of-concept demonstration, this paper shows that targeted adversarial attacks are achievable on black box models using straightforward methods.

Furthermore, the inclusion of momentum mutation and adding noise exclusively to high frequencies improved the effectiveness of our approach. Momentum mutation exaggerated the exploration at the beginning of the algorithm and annealed it at the end, emphasizing the benefits intended by combining genetic algorithms and gradient estimation. Restricting noise to the high frequency domain improved upon our similarity both subjectively by keeping it from interfering with human voice as well as objectively in our audio sample correlations. By combining all of these methods, we are able to achieve our top results.

In conclusion, we introduce a new domain for black box attacks, specifically on deep, nonlinear ASR systems that can output arbitrary length translations. Using a combination of existing and novel methods, we are able to exhibit the feasibility of our approach and open new doors for future research.

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
