# OpenReview forum: "Targeted Adversarial Examples for Black Box Audio Systems"
_ICLR.cc/2019/Conference_

### Official Review · AnonReviewer2 · 2018-11-01
**The paper is not well-positioned against the existing literature on black-box attack. Its empirical evaluation is somewhat sloppy.**

**Rating:** 3
**Confidence:** 4

**Review:**

PAPER SUMMARY:

This paper introduces a biologically motivated black-box attack algorithm.
The target model in this case is DNN applied to the ASR context (automatic speech recognition system).

NOVELTY & SIGNIFICANCE:

The proposed approach extends the previous genetic approach of (Alzantot et al., 2018) to attack a more complicated ASR system (that handles phrases and sentences). The new contribution here is an add-on momentum mutation component on top of the existing genetic programming architecture of (Alzantot et al., 2018) as illustrated in Figure 3.

This however appears very incremental seeing that integrating the mutation component into existing system is straight-forward and that mutation is not even a new concept -- it has always been a vital component in genetic programming paradigm.

It is also unclear how this mutation component improves over the existing work (more on this in the sections below).

Another issue is this work seems to ignore the recent literature on adversarial black-box attacks to DNN model. To list a few:

Chen, P.-Y.; Zhang, H.; Sharma, Y.; Yi, J.; and Hsieh, C.-J. 2017b.
ZOO: Zeroth-order optimization-based  black-box attacks to deepneural networks without training substitute models.
In Proceedings of the 10th ACM Workshop on Artificial Intelligence and Security (15-26) ACM

Cheng,  M.;  Le,  T.;  Chen,  P.-Y.;  Yi,  J.;  Zhang,  H.;  and  Hsieh,C.-J.2018.
Query-efficient hard-label black-box attack:  An optimization-based approach. arXiv preprint arXiv:1807.04457

While these works have not been used to attacking ASR system, they should be directly applicable to such system since after all, they are black-box attacks. I think the proposed method needs to be compared with these works.

TECHNICAL SOUNDNESS:

I find it surprising that even though the proposed method is claimed to be a black-box attack but in the end, it actually exploits the fact that the target model uses CTC decoder. This pertains specifically to the target model's internal architecture and a black-box attack is not supposed to know this.

CLARITY:

The paper is clearly written.

EMPIRICAL RESULTS:

I do not understand this statement:

"That 35% of random attacks were successful in this respect highlights the fact that black box
adversarial attacks are definitely possible and highly effective at the same time"

Why is 35% successful attack rate a positive result? The result tends to suggest that this is an attack with low success rate.

The 2nd paragraph in 3.2 seems to give a vague explanation: "the vast majority of failure cases are only a few edit distances away from the target.

This suggests that running the algorithm for a few more iterations could produce a higher success rate, although at the cost of correlation similarity".

Given the above statement, I do not see why the authors didn't actually "run the algorithm for a few more iterations" to verify it ...

I am also curious why is the success rate of the proposed method is significantly lower than that of the existing system -- I assume "single word black box" is the work of (Alzantot et al., 2018).

I find the empirical evaluation somewhat sloppy: why are the tested method not compared on the same benchmark? How do we interpret the results then?

REVIEW SUMMARY:

The paper misses the recent literature on black-box attack. The authors need to compare with those to demonstrate the efficiency of their proposed work. I also find the contribution of this paper too incremental & its empirical evaluation appears somewhat sloppy and not convincing (see my specific comments above).

---

> ### Author Response · Authors · 2018-11-12
> **Author response to reviewer 2**
>
> Dear reviewer,
>
> Thank you for your detailed comments on our paper and finding it well written.
>
> == Response to: "Another issue is this work seems to ignore the recent literature on adversarial black-box attacks to DNN model"
>
> Thank you for providing relevant literature. The first method provided, ZOO [1], is in fact closely related to finite gradient estimation, introduced in [2], which is the method our attack uses in phase 2. As for the second method provided, the paper introduces an attack for hard-label black box settings, where even output logits are not known, and where optimization is much more difficult [3]. In our setting, we assumed output logits are known, and so hard-label methods are not needed, as using output logits make optimization much easier.
>
> == Response to: "the proposed method is claimed to be a black-box attack but in the end, it actually exploits the fact that the target model uses CTC decoder"
>
> In the black-box setting, all that is required is access to the output logits of the model (as specified in Section 1.3). Any loss function that uses the output logits and target phrase could be applied to our method; we chose CTC loss as it is a well-known loss function suited for this task. Thus, the fact that both the training of the victim model and our attack use CTC loss is mostly coincidence.
>
> == Response to: "The result tends to suggest that this is an attack with low success rate"; "why is the attack success rate of the targeted attack success rate of the proposed method is significantly lower than that of the existing system"
>
> As stated in Section 1.2, the difficulty of this task comes in attempting to apply black-box optimization to a deeply-layered, highly nonlinear decoder model that has the ability to decode phrases of arbitrary length. We would like to clarify that there has not been an existing black box system for targeting the DeepSpeech model; the black box method in [4] attacks a lightweight classification model, where the model uses a softmax loss to classify between 50 words. The DeepSpeech model is much more complex, namely in that it can decode phrases of arbitrary length, and each output state (50 states per second) of the recurrent structure has a softmax layer, whereas in the classification model there was only one softmax.
>
> == Response to: "why are the tested method not compared on the same benchmark"
>
> In Table 1, we attempt to provide a standardized comparison of the previous techniques; naturally, there will be gaps since both are different attack types and attack different models. Our method is the first to extend black-box attacks to ASR systems; thus, we are aiming to be a baseline on this task, and there are no direct previous baselines to compare with. For example, datasets are different since DeepSpeech can accept any input, whereas the classification model can only accept 1-word phrases.
>
> References:
>
> Chen, P.-Y.; Zhang, H.; Sharma, Y.; Yi, J.; and Hsieh, C.-J. 2017b.
> ZOO: Zeroth-order optimization-based  black-box attacks to deepneural networks without training substitute models.
> In Proceedings of the 10th ACM Workshop on Artificial Intelligence and Security (15-26) ACM
>
> A. Nitin Bhagoji, W. He, B. Li, and D. Song. Exploring the Space of Black-box Attacks on Deep
> Neural Networks. ArXiv e-prints, December 2017.
>
> Cheng,  M.;  Le,  T.;  Chen,  P.-Y.;  Yi,  J.;  Zhang,  H.;  and  Hsieh,C.-J.2018.
> Query-efficient hard-label black-box attack:  An optimization-based approach. arXiv preprint arXiv:1807.04457
>
> M. Alzantot, B. Balaji, and M. Srivastava. Did you hear that? Adversarial Examples Against
> Automatic Speech Recognition. ArXiv e-prints, January 2018.

---

### Official Review · AnonReviewer1 · 2018-11-05
**Targeted adversarial examples for black box audio systems**

**Rating:** 6
**Confidence:** 4

**Review:**

In "Targeted adversarial examples for black box audio systems" the authors look at an adversarial problem in neural nets for audio processing. There is quite a lot of recent interest in adversarial problems in machine learning. That work is mostly on the image side, and so this work is very topical. The problem is to modify an audio signal without changing how it sounds to the human ear, so that it is interpreted as the attacker wishes by the neural network. In the black box approach, the weights of the neural network are not known by the attacker. The attacker however must be able to present modified audio and learn the network's interpretation as often as the attacker wants. This work is very exciting and topical, and of interest to the ICLR community.

The authors demonstrate a proof of concept using the recent DeepSpeech model, and they connect very well with recent literature on adversarial networks.

The particular algorithm the authors propose is based on genetic algorithms. I thought that this was a weak part of the paper, because genetic algorithms are quite ad hoc and have few theoretical guarantees when compared to SMC, MCMC, nested sampling or herding, which all do basically the same thing as genetic algorithms. This can lead to loose ends, such as the "momentum mutation" introduced by the authors in 2.2, wherein probability of mutation increases as the population fails to adapt. It is true that momentum mutation would avoid local maxima, but it would also take the solution away from global maxima through a sort of "sampling noise" (the global maxima is a point at which the population also "fails to adapt", as there's no more adaptation to be done). It's unclear if this is a problem, but things like annealed importance sampling also deal with the same problem (or effective sample size of SMC), and they have theory to back them up.

---

> ### Author Response · Authors · 2018-11-12
> **Author response to reviewer 1**
>
> Dear reviewer,
>
> Thank you for your detailed comments on our paper and finding it of interest.
>
> The suggestions of various black-box algorithms to use are appreciated, and could promise to generate higher quality adversarial examples. Such extensions would definitely be welcome in future work, as in this paper we attempt to establish a baseline that future methods can compare to.

---

### Official Review · AnonReviewer3 · 2018-11-07
**Evaluation is weak**

**Rating:** 4
**Confidence:** 3

**Review:**

This paper proposes a black-box attack on multi-word ASR systems.  Most work on black-box attacks have focused on tasks in vision. This work adds to the literature on attac
ks on speech systems. The key novelties are the handling of a loss function over multiple decodings as well as the use of novel genetic algorithms to generate the adversari
al examples.

A weakness of this paper is that they do not compare to the closely related Alzantot et al. work. While the latter is focused on single word settings and is thus solving an
 easier problem, what would happen if the Alzantot et al. method was applied to each


While the idea is interesting but incremental, the evaluation of the approach is weak.

1. Insted of choosing random pairs of words as target phrases, it would be interesting to pick phrases that are likely to occur in English and to ask how success rate varie
s as a function of the initial phrase and target phrase.

2. To confirm that the resulting adversarial examples are similar to audio samples in the original dataset, the authors should do user studies. This is a key component in e
valuating the efficacy of such attacks. The cross correlation is useful but does not get at perceptual similarity.

3. Table 1 is not useful since either the datasets are different or information is not given on the specific white box attacks.

4. Does increasing the iterations lead to a higher success rate as claimed at end of page 7?


Abstract:
1. This sentence is misleading : "Current work..are known" given the Alzantot et al. work focuses on black-box attacks.

---

> ### Author Response · Authors · 2018-11-12
> **Author response to reviewer 3**
>
> Dear reviewer,
>
> Thank you for your detailed comments on our paper.
>
> == Response to: "ask how success rate varies as a function of the initial phrase and target phrase"
>
> Thank you for this insight; we agree this would be useful to see how the attack performs for phrases likely in the general English language.
>
> == Response to: "the authors should do user studies"
>
> We agree that user studies would most effectively verify the efficacy of the attack; however, this would incur significant costs for the authors. In lieu of a human study, the Carlini & Wagner attack measured attack via cross-correlation, and so we use the measure for similarity.
>
> == Response to: "Table 1 is not useful since either the datasets are different or information is not given on the specific white box attacks."
>
> In Table 1, we attempt to provide a standardized comparison of the previous techniques; naturally, there will be gaps since both are different attack types and attack different models. Since our method is the first to extend black-box attacks to ASR systems, there are no direct previous baselines to compare with. For example, datasets are different since DeepSpeech can accept any input, whereas the classification model can only accept 1-word phrases from the predefined set of classes.
>
> == Response to: "Does increasing the iterations lead to a higher success rate as claimed at end of page 7?"
>
> Yes it does; we will change the wording to make it less ambiguous and make sure to add a couple extra figures in the final version as verification.

---

### Meta-Review · Area_Chair1 · 2018-12-14
**Limited novelty compared to previous works**

**Confidence:** 5
**Recommendation:** Reject

**Metareview:**

The authors propose an algorithm for generating adversarial examples for ASR systems treating them as black boxes.

Strengths
- One of the early works to demonstrate black box attacks on ASR system that recognize phrases instead of isolated words.

Weaknesses
- The approach assumes that the logits are available, which may not be realistic for most ASR systems when they are used in practice -- typically only the final transcription is available.
- Although the technique is applied to continuous speech, algorithmic improvements over prior work of Alzanot et al. is minimal.
- Evaluation is weak. For example, cross correlation cannot completely capture the adversarial nature of a generated audio sample.
- The authors use a genetic algorithm for generating new set of examples which are pruned and mutated. It’s not clear what guarantees exist that the algorithm will eventually succeed.

The reviewers agree that the presented work puts forth an interesting research direction. But given the deficiencies of the current submission as pointed out by the reviewers, the recommendation is to reject the paper.